# Clinical Considerations for a Family with Dilated Cardiomyopathy, Sudden Cardiac Death, and a Novel *TTN* Frameshift Mutation

**DOI:** 10.3390/ijms22020670

**Published:** 2021-01-12

**Authors:** Emanuele Micaglio, Michelle M. Monasky, Andrea Bernardini, Valerio Mecarocci, Valeria Borrelli, Giuseppe Ciconte, Emanuela T. Locati, Marco Piccoli, Andrea Ghiroldi, Luigi Anastasia, Carlo Pappone

**Affiliations:** 1Department of Arrhythmology, IRCCS Policlinico San Donato, Piazza Edmondo Malan 1, San Donato Milanese, 20097 Milan, Italy; Emanuele.micaglio@grupposandonato.it (E.M.); michelle.monasky@grupposandonato.it (M.M.M.); Andrea.Bernardini@grupposandonato.it (A.B.); valerio.mecarocci@gmail.com (V.M.); valiborrelli91@gmail.com (V.B.); g.ciconte@gmail.com (G.C.); EmanuelaTeresina.Locati@grupposandonato.it (E.T.L.); 2Laboratory of Stem Cells for Tissue Engineering, IRCCS Policlinico San Donato, San Donato Milanese, 20097 Milan, Italy; piccolimarco83@gmail.com (M.P.); andrea.ghiroldi@gmail.com (A.G.); Anastasia.luigi@hsr.it (L.A.); 3Vita-Salute San Raffaele University, 20132 Milan, Italy

**Keywords:** dilated cardiomyopathy, *TTN*, titin, sudden cardiac death, genetic testing, mutation, humans, family, deletion, truncating

## Abstract

Dilated cardiomyopathy (DCM) is the leading indication for heart transplantation. *TTN* gene truncating mutations account for about 25% of familial DCM cases and for 18% of sporadic DCM cases. The clinical relevance of specific variants in *TTN* has been difficult to determine because of the sheer size of the protein for which *TTN* encodes, as well as existing extensive genetic variation. Clinicians should communicate novel clinically-relevant variants and genotype–phenotype associations, so that animal studies evaluating the molecular mechanisms are always conducted with a focus on clinical significance. In the present study, we report for the first time the novel truncating heterozygous variant NM_001256850.1:c.72777_72783del (p.Phe24259Leufs*51) in the *TTN* gene and its association with DCM in a family with sudden death. This variant occurs in the A-band region of the sarcomere, in a known mutational hotspot of the gene. Truncating titin variants that occur in this region are the most common cause of DCM and have been rarely reported in asymptomatic individuals, differently from other pathogenic *TTN* gene variants. Further studies are warranted to better understand this particular clinically-relevant variant.

## 1. Background

Dilated cardiomyopathy (DCM) is the leading indication for heart transplantation [1], characterized by left ventricular dilatation, increased left ventricular volume, and decreased ejection fraction [2], affecting approximately 1:250 individuals [1]. More than 35% of patients clinically affected by DCM harbor either a heterozygous or homozygous mutation in at least one of several genes, encoding for sarcomeric, cytoskeletal, or nuclear proteins [3]. The percentage of molecularly confirmed cases is today not easily assessed, mainly because of phenotypic overlapping, but has recently been reported to be as high as 69% in whole exome studies [4]. In spite of this, the average clinical sensitivity of genetic testing for DCM, using next-generation sequencing (NGS), can be estimated to be much lower, in particular around 37% [5]. Furthermore, the rate of variants of uncertain significance found in DCM patients has been reported to be about 28% [6].

A wide genetic heterogeneity exists in DCM [7], with variants in at least 39 genes contributing to the phenotype [8]. However, some genotype–phenotype correlations for DCM patients are well known [9], and the severity can depend upon both the gene in which the variant is located and the kind of mutation [10]. For instance, truncating mutations in the *TTN* gene account for about 25% of familial DCM cases and for 18% of sporadic DCM cases [11].

*TTN* is one of the largest human genes, composed of 363 exons [12] encoding the largest human protein, a very important component of both cardiac and skeletal muscle. The clinical relevance of specific variants in *TTN* has been difficult to determine because of the sheer size of the protein as well as extensive genetic variation that exists [13]. Basic science using animal research has made great strides in our understanding of titin. For example, a recent study described the role of titin in active force production and how titin can influence force production at various diastolic calcium levels [14]. Maintaining a bridge between basic science and the clinic is of vital importance to the advancement of clinically-relevant research. Clinicians should strive to communicate the novel variants that they find clinically relevant, reporting novel genotype–phenotype associations, so that animal studies evaluating the molecular mechanisms can continue to always be conducted with a focus on clinical significance.

In the present study, we report for the first time the novel truncating heterozygous variant NM_001256850.1:c.72777_72783del (p.Phe24259Leufs*51) in the *TTN* gene and its association with a familial form of DCM in a family with a history of sudden death.

## 2. Case Presentation

The proband is a 46-year-old man with a height of 174 cm, a weight of 74 kg, and a body mass index (BMI) of 24.44 kg/m^2^. At the age of about 45 years old, due to the sudden onset of shortened breath with abdominal bloating and precordial pain, the proband performed a cardiology outpatient clinical evaluation. Because of both his family history of sudden death and the clinical presentation, the proband performed an electrocardiogram (ECG), showing an incomplete left bundle branch block (LBBB) and a first-line laboratory work up, with normal values, including normal plasmatic transaminases (AST = 19 U/L, ALT = 21 U/L, gamma GT = 18 U/L). The abdominal ultrasound revealed an enlarged liver, so a further biochemical workup was performed. Remarkably, the plasmatic dosage of both LDL cholesterol and *N*-terminal fragment of pro B-natriuretic peptide (NT pro-BNP) was altered. In particular, LDL cholesterol was 138 mg/dL (optimal values less than 100 mg/dL) and NT pro-BNP was 706 pmol/L (normal values less than 125 pmol/L at the proband’s age). Therefore, to rule out ischemic cardiomyopathy, the proband underwent a coronary arteriography elsewhere, which was normal. Then, the proband was started on therapy with Acetylsalicylic Acid 100 mg per day, Amiodarone 200 mg per day, Furosemide 50 mg per day, Potassium Kanrenoate 50 mg per day, Bisoprolol 2.5 mg per day, Omeprazole 20 mg per day, and Potassium Chloride Retard 600 mg per day. Next, he performed echocardiography, also elsewhere, showing an ejection fraction of 25% (normal values 64–84% according to sex, BMI, and age), a stroke volume of 87.2 mL (normal values 59.0–93.0 mL), a cardiac output of 7.1 L/min (normal values 4.8–7.5 L/min), and a stroke index of 51 mL/m^2^ (normal values 30–48 mL/m^2^). These results prompted the execution of heart magnetic resonance imaging (MRI), which reports as follows: “globular morphology of the left ventricle” with an end diastolic volume of 348 mL (normal values 83–138 mL); thickness of interventricular wall at the end of diastole of 11 mm (normal values = up to 14 mm); and diffuse ventricular hypokinesia, with severe depression of ejection fraction (=25%). In addition, the cardiac magnetic resonance imaging (MRI) identified a streak of late enhancement of contrast in both the medial and basal regions of the interventricular wall. Diffused increasing of extracellular volume (ECV) values was consistent with diffuse fibrosis (33%, normal value around 25%). There were also signs of compression in the right ventricle due to the mass effect of the dilated left ventricle. 

At the subsequent clinical re-evaluation with the aforementioned imaging, the proband received the diagnosis of DCM, noting especially a high blood pressure (170/100 mmHg measured in clinostatism in the left arm) and a marked reduction in the ejection fraction (EF), lower than 20%, within a couple of months after the MRI and after full adherence to the prescribed therapy. This prompted a change in the drug therapy, adding the combination of Sacubitril + Valsartan (45 mg + 51 mg per day, respectively) to the aforementioned therapy. A subsequent arrhythmologic evaluation resulted in the recommendation for a cardiac resynchronisation therapy defibrillator (CRT-D). At the last cardiologic evaluation, the ejection fraction measured at cardiac transthoracic ultrasound was 44% and the plasmatic NT pro-BNP value was 27 pmol/L (normal values less than 125 pmol/L at the proband’s age), confirming the very good outcome resulting from the therapeutic changes, which also included normal blood pressure (120/80 mmHg measured in clinostatism in the left arm) and concentrations of plasmatic ion sodium, potassium, calcium, and chloride. The proband’s mother had been diagnosed with DCM around the age of 30 years old, prior to which she had carried and given birth to two children. The mother then suddenly died at the age of 33 years and the autopsy performed elsewhere confirmed the significantly increased volume of the left ventricle (values unknown because of the lack of clinical documentation).

Echocardiographic examination repeated in our facility in the proband demonstrated a dilated left ventricle and a suspicion for left ventricular noncompaction (Figure 1). Incomplete left bundle branch block (LBBB) was seen on the electrocardiogram (ECG) (Figure 2). 

Genetic analysis by NGS was performed on the following gene panel from genomic DNA extracted from peripheral blood with a medium coverage of 106 X: *ABCC9, ACTC1, ACTN2, ANKRD1, BAG3, CAV3, CRYAB, CSRP3, DES, DMD, DNAJC19, DOLK, DSC2, DSG2, DSP, EMD, EYA4, FHL1, FHL2, FKRP, FKTN, FLNC, GATAD1, GLA, ILK, JUP, LAMA4, LAMP2, LDB3, LMNA, MYBPC3, MYH6, MYH7, MYL2, MYL3, MYPN, NEXN, PDLIM3, PKP2, PLN, PRKAG2, RAF1, RBM20, RYR2, SCN5A, SDHA, SGCD, TAZ, TBX20, TCAP, TMEM43, TMPO, TNNC1, TNNI3, TNNT2, TPM1, TTN, TTR, TXNRD2, VCL*. We fully analyzed the whole *TTN* gene, including PEVK domains, with a medium coverage of 106X.

After this genetic testing, only the heterozygous variant NM_001256850.1:c.72777_72783del (p.Phe24259Leufs*51) in the *TTN* gene (Leiden Open Variation Database: https://databases.lovd.nl/shared/individuals/00314916) was found, and this result was confirmed by Sanger Sequencing (Figure 3).

### 2.1. Assessment of Family Members

The family pedigree can be seen in Figure 4. The proband’s 48-year-old sister underwent an unremarkable ECG MRI elsewhere. When she came to our center, she underwent an unremarkable echocardiography. Genetic testing was performed from genomic DNA extracted from peripheral blood by Sanger sequencing to search for the NM_001256850.1:c.72777_72783del (p.Phe24259Leufs*51) variant in the *TTN* gene, which resulted in a finding of the absence of the variant.

The proband’s father (72 years old) was tested elsewhere for the familial variant NM_001256850.1:c.72777_72783del (p.Phe24259Leufs*51) in the *TTN* gene from genomic DNA extracted from peripheral blood. The proband reported to us the absence of this variant in his father, who also performed an unremarkable 12-lead ECG and cardiac ultrasound examination. This suggests that the variant in the *TTN* gene found in the proband was maternally inherited. It is also possible that the variant found in the proband is a *de novo* variant that was inherited from neither parent. However, it is more probable that the variant was maternally inherited.

The proband’s 12-year-old son underwent an unremarkable ECG and echocardiography elsewhere. The proband’s 16-year-old niece and 15-year-old nephew did the same with the same unremarkable results. Genetic testing was performed in these three patients from genomic DNA extracted from peripheral blood by Sanger sequencing to search for the NM_001256850.1:c.72777_72783del (p.Phe24259Leufs*51) variant in the *TTN* gene, which resulted in a finding of the absence of the variant.

### 2.2. In Silico Predictions

The c.72777_72783del variant was classified as pathogenic according to the American College of Medical Genetics (ACMG) criteria [16,17]:PVS1 (pathogenic very strong): Null variant (frame-shift) affecting gene *TTN*, which is a known mechanism of disease (gene has 1181 known pathogenic variants, which is greater than minimum of 3), associated with cardiomyopathy, familial hypertrophic 9 and cardiomyopathy, dilated, 1G.PM2 (pathogenic moderate): Variant not found in gnomAD exomes (good gnomAD exomes coverage = 75.8). Variant not found in gnomAD genomes (good gnomAD genomes coverage = 32.4).PP3 (pathogenic supporting): Pathogenic computational verdict based on one pathogenic prediction from genomic evolutionary rate profiling (GERP) versus no benign predictions.

## 3. Discussion

Truncating mutations in the gene encoding for the titin protein are the most common cause of inherited DCM [18,19,20] and can result in a quite severe form of DCM, including frequent arrhythmia [19] and even heart transplantation [20]. However, many truncating variants in the *TTN* gene are found in the general population, even in healthy controls, calling into question the clinical significance of those variants [1]. Thus, for genetic testing to be useful, it is imperative to understand the clinical significance of specific variants [1]. In the present study, we report for the first time the novel truncating heterozygous variant NM_001256850.1:c.72777_72783del (p.Phe24259Leufs*51) in the *TTN* gene and its association with a familial form of DCM in a family with a history of sudden death.

The novel variant described herein occurs in the A-band region of the cardiac sarcomere. Truncating titin variants that occur in this region are the most common cause of DCM [21]. However, the presence of variants of this type in the general population can create confusion for the interpretation of their significance [21]. According to the Varsome database [17], the frequency in the general population of this novel variant identified in the present study was not detected. This could be interpreted as the frequency of this variant being incredibly low—estimated as much less than 0.001% in the general population. This evidence further suggests a pathogenic effect of this particular variant, according to current knowledge (1000 genomes, Exome Aggregation Consortium (ExAc) and Single Nucleotide Polymorphism Database (DB-SNP)).

The evolutionary conservation of a particular gene is another factor to consider when determining the significance of a particular variant. Generally speaking, the more conserved the gene, the more likely a variant would result in a pathogenic effect. According to the Varsome website [17], the GERP score is used to determine this, ranging from −12.3 to 6.17, with 6.17 being the most conserved. The median GERP score for the c.72777_72783del variant is listed as 6.03 [17], predicting a pathogenic effect resulting from a variant in this highly evolutionarily conserved region.

The importance of titin comes from its function connecting the Z disk to the M line in the sarcomere, providing the passive elasticity and resting tension of the muscle, and contributing to force transmission and signaling [22,23]. Its functions affect a great number of proteins, including proteins at the Z disk, I band, and M line, as well as proteins involved in post-translational regulation. Variants in the *TTN* gene encoding a defective titin protein could greatly affect muscle stiffness, resulting in the activation of compensatory pathways to preserve the cardiac output. However, these compensatory mechanisms, when sustained, eventually lead to a deterioration of the intended signaling process in the cardiomyocyte. 

In this study, the proband’s father, sister, son, niece, and nephew were found to be negative for the familial variant and have a completely normal clinical picture. The majority of DCM is inherited in an autosomal dominant manner and has an age-dependent penetrance. This is also true for DCM cases caused by heterozygous titin mutations, which have been reported to have a penetrance of more than 95% in people older than 40 years [11]. A more recent study assessed the penetrance of *TTN* truncating mutations as very high, reaching 100% by the age of 70 years [9]. Therefore, it appears that neither the proband’s 72-year-old father nor the proband’s 48-year-old sister will develop DCM.

Taken together, these data strongly suggest that the heterozygous state for c.72777_72783del (p.Phe24259Leufs*51) in the *TTN* gene is a novel cause of DCM. After careful genetic counselling, and based on the absence of mutations in the other genes associated with DCM, the recurrence risk appears to be 50%.

This study provides crucial human data upon which future studies, such as functional studies in animal models, can be based. These data describe a novel variant, never before described, in a family with DCM, when it is known that variants in the *TTN* gene can sometimes be responsible for other pathologies, such as hypertrophic cardiomyopathy, arrhythmogenic right ventricular cardiomyopathy, or restrictive cardiomyopathy, while some variants in the *TTN* gene may not even be pathogenic at all [13]. Additionally, the *in silico* studies presented above predict a pathogenic effect. While animal models can be useful to expand our understanding of specific variants, creating a transgenic mouse model, for example, is very expensive, time-consuming, and exhausting, and so it is important for physicians to communicate with basic scientists the variants they feel are important and their hypotheses about any causative effects. Human data is invaluable in the development of animal models, and thus this clinical data is meant to provide the stepping-stone for future functional studies, which could be performed by basic scientists, who have different skills and resources. With this said, differences between species exist; therefore, even in the presence of functional studies in mice or other species, human data is still precious [24]. Thus, we communicate here the presence of the variant NM_001256850.1:c.72777_72783del (p.Phe24259Leufs*51) in the *TTN* gene that occurs in a family with DCM and sudden death, in the absence of other detected variants, suggesting a severe phenotype resulting from this variant. Moreover, the proband’s clinical picture encompasses left ventricular noncompaction (LVNC) as well. The combined therapy with Sacubitril + Valsartan was able to improve the proband’s heart function, as already described in other patients clinically affected by DCM [25]. It is known that heterozygous *TTN* variants can be the cause of LVNC, in one study accounting for 19% of all LVNC cases, caused mainly by truncating *TTN* variants [26]. This could be related to the fact that titin is needed for sarcomere assembly and force transmission [27], contributing meaningfully to both systolic and diastolic function of the heart. However, the mechanisms leading to LVNC in the presence of *TTN* truncations are still unclear [28], and could be the subject of future studies.

## 4. Concluding Remarks

The novel truncating heterozygous variant NM_001256850.1:c.72777_72783del (p.Phe24259Leufs*51) in the *TTN* gene appears to be the cause of DCM and sudden death in the family presented, suggesting a pathogenic role for this variant and providing the clinical data on which basic science studies can be based to further understand this particular clinically-relevant variant.

## Figures and Tables

**Figure 1 ijms-22-00670-f001:**
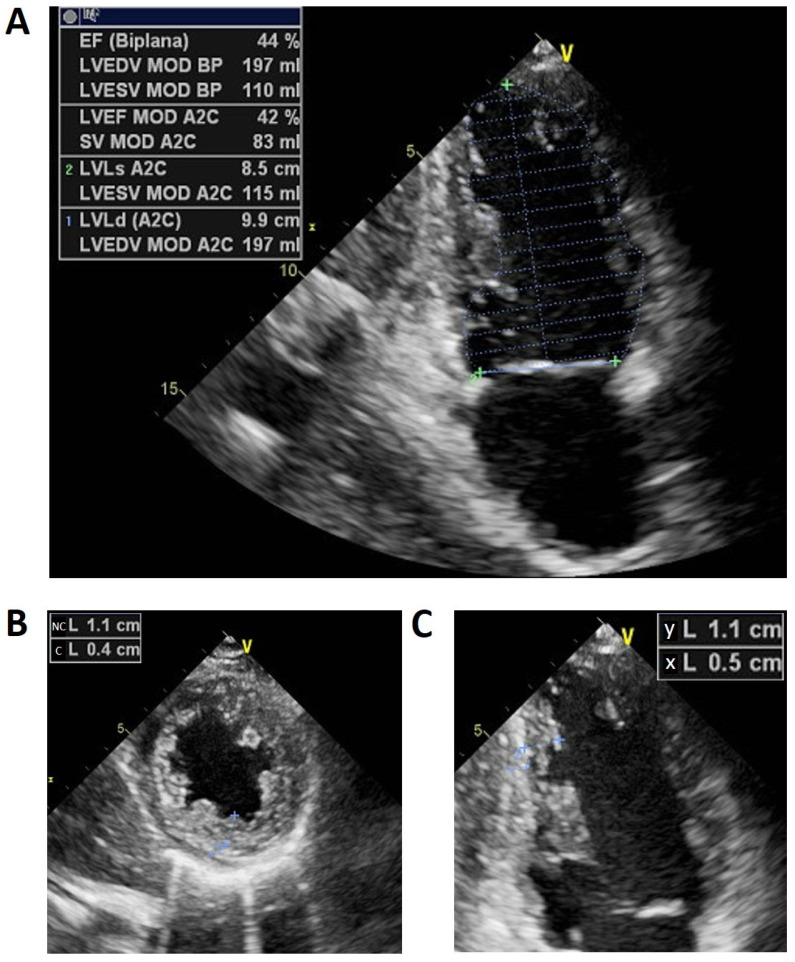
(**A**) Apical two-chamber view showing left ventricle (LV) dilatation (end-diastolic volume indexed 105 mL/m^2^ and end-systolic volume indexed 58 mL/m^2^), with a reduced ejection fraction (EF) (44%, biplane). (**B**) Parasternal short axis of LV during the systole phase showing a compact epicardial layer (C) and noncompacted endocardial layer (NC) mainly in the mid apical inferior and inferolateral left ventricle with prominent trabeculations and a maximal end systolic NC/C trabecular ratio > 2 (criteria reference: [15]). (**C**) Ratio of the distance from the epicardial surface to the trough of the trabecular recess (X) to the distance from the epicardial surface to the peak of the trabeculation (Y) with a ratio ≤0.5.

**Figure 2 ijms-22-00670-f002:**
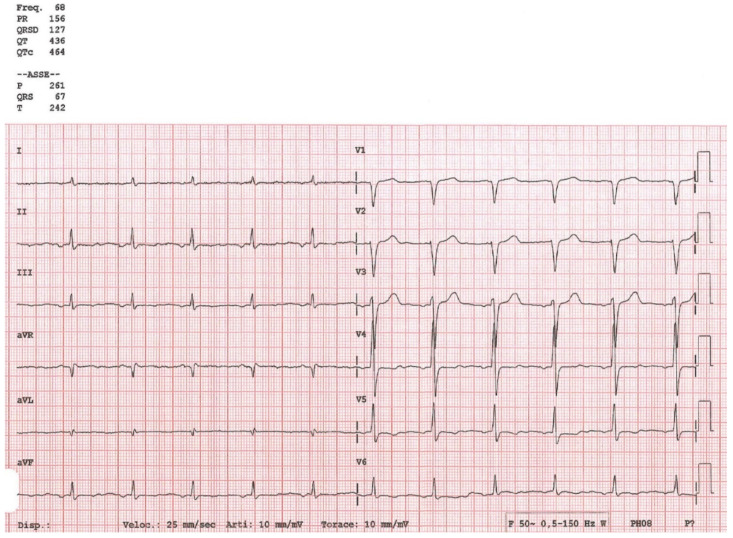
Electrocardiogram of the proband showing incomplete left bundle branch block.

**Figure 3 ijms-22-00670-f003:**
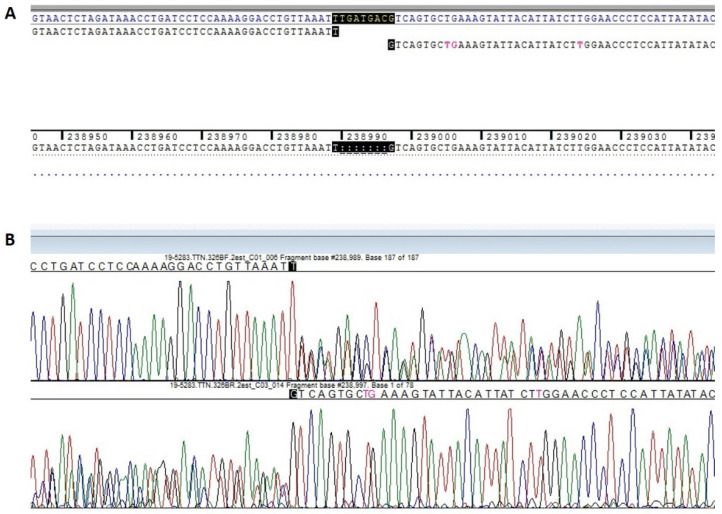
Post-next-generation sequencing (NGS) Sanger confirmation of the heterozygous *TTN* variant NM_001256850.1: c.72777_72783del. (**A**) Deletion of seven nucleotides in the DNA fragment of the *TTN* gene. (**B**) Demonstration of frameshift in the proband’s DNA fragments in the *TTN* gene mutated region.

**Figure 4 ijms-22-00670-f004:**
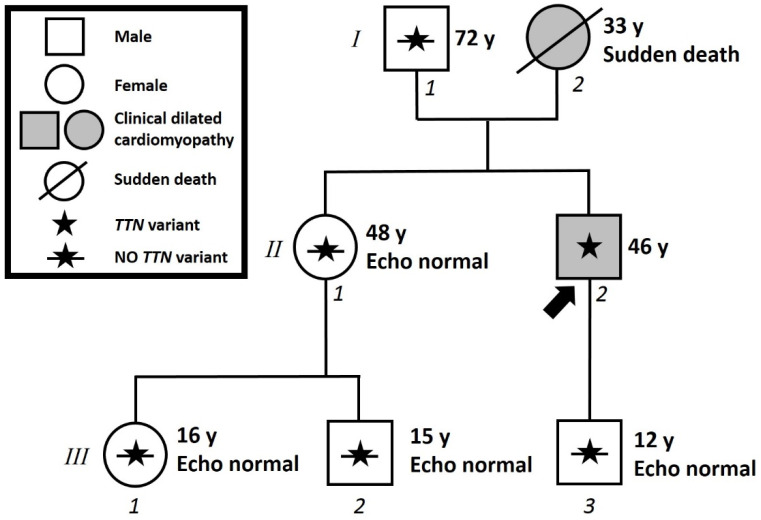
**Family pedigree.** Proband identified with arrow. Square: male; circle: female; shaded: clinically affected by dilated cardiomyopathy; star: molecularly confirmed *TTN* mutation; star with slash: genetically tested and negative for *TTN* mutation; y = years old at diagnosis; Echo: echocardiography.

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
