# Peer review of "Clinical Considerations for a Family with Dilated Cardiomyopathy, Sudden Cardiac Death, and a Novel TTN Frameshift Mutation"

_ijms, 2021, doi:10.3390/ijms22020670_

Round 1

Reviewer 1 Report

After reading the revised version of the manuscript, I realized that there are still some points not addressed.

  • I have severe doubts if this manuscript should be really published in 'International Journal of Molecular Sciences'. Most of the presented data are clinical data and structural or molecular data are completely missing. Therefore, I suggest to submit this manuscript to a sister journal like 'Journal of Clinical Medicine' or ‘Genes’. A special issue about cardiovascular genetics is currently available at Genes ... In my view this manuscript fits better to these two journals. https://www.mdpi.com/journal/genes/special_issues/Cardiovascular_Genetics.
  • Structural data are still not presented (comment 4 of my last review report). This point was completely ignored by the authors. Why? The response of the authors is completely not addressing this point in an adequate way.
  • In addition my comments about haploinsufficiency were completely ignored (comment 5). The authors are right, that beside haploinsufficiency further putative mechanisms like protein-protein interactions might be involved. However, where are data about these points??? To do nothing and to leave the pathomechanisms completely open is not acceptable for a manuscript submitted to a molecular journal.
  • The authors mention in their response that 1,181 known pathogenic variants in TTN have been published. What sense does it make to publish an additional mutation without presenting any functional data or presenting a cosegregation in a large family? This argumentation is strange. Nearly every cardiovascular genetic lab has found several novel TTN mutations in DCM patients. So why is it really relevant to publish this additional TTN variant as number #1,1182? A simple entry in relevant databases would be also possible as an alternative to a full article. You definitely need functional / structural data to justify publication of this TTN variant. 

Good luck with the revision!

Author Response

Reviewer 1:

We would like to thank the reviewer for his/her important comments.

We appreciate the reviewer pointing out alternative options for submission, such as to the 'Journal of Clinical Medicine' or ‘Genes’, and for pointing out the existence of the special issue about cardiovascular genetics currently available at ‘Genes’. We appreciate to be informed about these options. At the same time, we also feel that discussing a molecular change in the DNA and its association for the first time ever with the clinical picture of DCM, in a family with sudden death, and presenting echocardiography and electrocardiogram of the patient is important, given that this is the first step in understanding this variant.

Importantly, we think that one thing that should also be noted is the presence of a possible overlap syndrome. We are aware of large amounts of data regarding dilative cardiomyopathies associated with TTN truncating mutations. Before submitting to IJMS, we checked whether other cases of overlapping syndrome due to TTN truncating mutations existed already, and we did not find anything relating to DCM + LVNC. Thus, the phenotype appears to be more complex than just DCM, perhaps also contributing to an overlap syndrome between DCM and LVNC, in the presence of familial history of sudden death. This is explained better in the revision.

Functional studies done in translational medicine should focus on the benefit to the clinic and ultimately patient treatment. Thus, it is important for clinicians to communicate to the basic scientists odd or unusual clinical findings. Phenotypes can be different in animal models, in which the physiology is not always the same as in humans. There are healthy human carriers of TTN variants. Functional studies done on cells or in animal models, such as transgenic mice, have several limitations, and the results must always be verified in humans. Thus, ultimately, human data is the most important, especially for variants found in genes that have been associated with a number of different pathologies.

Additionally, usually, the groups that visit the patients are not the same groups that do the functional studies on cells or animal models. A lack of communication between various groups can hinder the advancement of science. Thus, it is always better to error on the side of ensuring communication.

The family presented now includes seven individuals. The five individuals who tested negative for the variant do not show any signs of DCM. The variant was not inherited from the proband’s father, which means that it is likely that the variant was inherited from the proband’s mother, who died suddenly at the age of 33 years.

Regarding the other comments, we have written in the discussion that this variant occurs in the A-band region of the sarcomere, in a known mutational hotspot of the gene, and that truncating titin variants that occur in this region are the most common cause of DCM. We are not sure exactly what further data the reviewer is asking for. At this time, RNA and/or protein data is not available. Instead, our goal was to communicate for the first time the clinical phenotype associated with this particular novel variant, provide family history (including ECG and echo), electrocardiogram and echocardiogram in the proband, and to provide the first and currently only available crucial human data to begin to understand this variant. Future studies exploring the functional effects of this variant may use this information as a stepping stone in cell and animal models, if so desired.

We thank the reviewer again for his/her important comments. We really appreciate his/her time in reviewing our manuscript.

Reviewer 2 Report

The corrections provided by the authors are sufficient to accept the paper. The paper can be published according the revised version

Author Response

We thank you for your time and your review.

Reviewer 3 Report

Micaglio et al. Presented a family with one affected DCM patient who had a positive family hisotry of sudden death at age of 33y.

They found a novel pathogenic frameshift mutation in TTN gene in the one affected person only and show clinical data on him as well extensively on his family.

Introduction is nicely written however case presentation is too long and poorly organized.  For example, there is no clinical presentation in terms of functional status,  presence of heart failure symptoms etc . Later description of standard 12-lead ECG should follow, and ECHO, Holter 24-hour ECG monitoring  instead of consideration of cardiac resynchonization and ICD placement.

Only then therapeutic considerations.

Also the family description is too long and in particular pictures of normal ECG and normal ECHO are useless and should be removed. They do not add anything to the whole case description.

In methodology (gene names) should be written in italic.

In Discussion part some consideration on TTN tv truncating variants in connection with LVNC should be placed.

Round 2

Reviewer 1 Report

Congratulations!

This manuscript is a resubmission of an earlier submission. The following is a list of the peer review reports and author responses from that submission.

Round 1

Reviewer 1 Report

The paper is not particularly interesting and does not report innovative data.

The paper of Roberts et al (Sci Transl Med. 2015;7) reported clear criteria to estimate the probability of pathogenicity of TTN truncation variants, already in 2015.

Furthermore, the paper loses segregation data (co-segregation should be proven over more than five meiosis), nor shows data with RNA or protein analyses.

Finally, carriers of TTN truncation mutations generally present with a milder cardiac phenotype. The case reported a DCM family, in which the proband (46 yo) showed a “severe depression of ejection fraction” and his mother suddenly died at the age of 33 years. Maybe, this clinical presentation is not mild, and would have required at least a comment.

Reviewer 2 Report

The manuscript by Micaglio E. et al describes the novel truncating heterozygous variant and its role in cardiac dysfunction by a case report. The data is well presented and results are adequately described. Overall, I found the manuscript is well written and interest to the readers.

Reviewer 3 Report

I have reviewed the manuscript 'Novel TTN p.Phe24259Leufs*51 Deletion Mutation Segregation in a Family with Dilated Cardiomyopathy' submitted by Micaglio et al. to International Journal of Molecular Scieces. The manuscript has several severe limations:

1.) The title of the manuscript is misleading, because a co-segregation analysis was not performed, because of lack of maternal genomic DNA.

2.) Evidence for a classification as a pathogenic TTN mutation is weak. No functional in vitro or in vivo data are presented. Only genetic and clinical information are presented.

3.) Please list all variants, which were found by NGS panel sequecing. How was filtering done with these additional variants?

4.) Please present structural information to the titin domain, where the mutation is located.

5.) Is haploinsufficiency caused by this mutation? Please present RNA and/or protein data.

Minor points:

Please explain all abbreviations